# OpenReview forum: "EVALUESTEER: Measuring Reward Model Steerability Towards Values and Preferences"
_ICLR.cc/2026/Conference — ICLR 2026 Conference Withdrawn Submission_

### Official Review · Reviewer_gLY2 · 2025-10-31

**Soundness:** 2
**Presentation:** 3
**Contribution:** 2
**Rating:** 2
**Confidence:** 3

**Summary:**

This paper introduces a benchmark dataset to evaluate RM's steerability against user value and stylistic preferences. The authors evaluated a set of LLM judges based RMs and found that they manifested noticeable gaps against Oracle labels.

**Strengths:**

* This paper studies an important problem on LLM alignment against user value and stylistic preferences.

* The construction of user profiles and values draw from real human data from World Values Survey.

* A diverse range of reward models are evaluated (based on models of varied sizes/capacity).

**Weaknesses:**

* When it comes to user value and preference assessments, having human in the loop is critical and necessary. The dataset construction has no human involvement, and it's unclear to what extent the generated pairs align with what people would prefer in practice. The fact that the authors use GPT-4o filter as the "oracle" basically caps the performance of human preference alignment by 4o's inherent capabilities. Having human validation will make the datasets much more credible and convincing.

* The main results presented by the paper (Figure 2) are confounded by the fact that 4o is the oracle. It is not surprising that models smaller than 4o didn't match the validation by 4o. This is less about human value alignment but more about aligning against the judgements made by 4o.

* The evaluation was only done on prompting-based LLM judges. When it comes to RM, it's important to test whether the introduced dataset can be used to improve RM through finetuning.

**Questions:**

Please see points in weaknesses

---

### Official Review · Reviewer_wP4b · 2025-11-01

**Soundness:** 2
**Presentation:** 2
**Contribution:** 2
**Rating:** 4
**Confidence:** 4

**Summary:**

The paper introduces EVALUESTEER, a synthetic benchmark probing whether LLM-based reward models can steer to user value and stylistic profiles. Using WVS-derived value statements and controlled style transforms, the authors evaluate six RMs across 11 prompting setups, finding style-over-substance biases and ≤75% accuracy with full context.

**Strengths:**

1. Steerability to pluralistic values and styles is underexplored for reward models; the paper targets an important gap beyond generic reward benchmarks.

2. Orthogonal manipulation of four value dimensions and four stylistic families with six pairwise comparison regimes is well designed for isolating effects.

3. Large evaluation (165,888 pairs), explicit prompting conditions (11), and an “oracle” filtration step make the pipeline easy to reason about; human validation adds rigor.

**Weaknesses:**

1. Completions are generated with GPT-4o and filtered by a GPT-4.1 oracle, while GPT-4.1-Mini is among evaluated judges. This tight coupling risks self-agreement bias and overestimates generalization. Suggestions: Use a disjoint oracle family (e.g., non-OpenAI) or a committee oracle with disagreement filtering; report results when the example pool is filtered by each of {OpenAI, Google, Meta, Alibaba} or by majority vote across families.

---

2. Even though you include a 4-vs-4 ablation, the main condition gives ≈200 value statements vs 4 style statements. This still encourages attention to surface style cues.
Suggestions: (1) Add a profile-summarization condition: top-k relevant value sentences via retrieval; (2) a matched-token-budget condition; (3) require RMs to cite which profile sentences justify their choice to check relevance sensitivity.


---


3. While the synthetic design isolates factors, it may not reflect messy real conversations where values are implicit and style/value signals co-occur with other attributes (domain knowledge, safety, politeness norms).
Suggestions: Include a human-in-the-loop subset: have annotators with known WVS-like profiles express preferences over the same pairs; report agreement and calibration vs EVALUESTEER labels.


---

4. “Readability,” “verbosity,” and “confidence” can correlate (longer → harder; confident → fewer hedges → sometimes simpler). Current stylometrics are univariate proxies.
Suggestions: Provide multivariate separability analyses (e.g., logistic regression predicting each style label using all stylometric features), and release per-sample metrics distributions showing minimal cross-loading.

**Questions:**

See Weakness Part

---

### Official Review · Reviewer_hFaN · 2025-11-08

**Soundness:** 2
**Presentation:** 3
**Contribution:** 2
**Rating:** 2
**Confidence:** 3

**Summary:**

The paper introduces EVALUESTEER, a benchmark designed to evaluate reward model (RM) steerability towards users’ value and style preferences. Built on the World Values Survey (WVS) and enriched with synthetic stylistic variations, EVALUESTEER tests how well reward models align model outputs with user profiles reflecting diverse human values and communication styles.

The synthesized benchmark comprises 165,888 preference pairs spanning four value dimensions (traditional, secular-rational, survival, and self-expression) and four stylistic dimensions (verbosity, readability, confidence, and warmth). Six RMs, including GPT-4.1-Mini and Skywork variants, are evaluated under 11 prompting conditions.

Key findings include:
  * Even with full context, top models achieve only ~75% accuracy, leaving a 25% gap to the oracle setting.
  * RMs show secular and self-expression biases on the Inglehart–Welzel value map.
  * Most RMs exhibit stylistic bias toward verbosity and confidence.
  * When values and styles conflict, RMs favor style over values ("style over substance" bias).

**Strengths:**

1. Overall, the paper is easy to follow, and the findings are clearly presented.

2. EVALUESTEER fills a research gap by systematically measuring RM steerability to user-specific values and styles.

**Weaknesses:**

The main weakness of the paper lies in its quality, especially the synthesized benchmark and the evaluated model sizes. To support the claims made in the paper, a manual check and more evaluations across different model sizes are needed.

1. [Quality] The samples in the benchmark are synthesized by GPT-4o, which raises concerns about their quality. For example, it is possible that GPT-4o may produce similar responses for opposites of certain types of WVS statements. In that case, the small gap in different values would not be caused by the inherent abilities of the evaluated models but by the benchmark itself. I strongly recommend conducting a manual check of the proposed benchmark, and if that is too difficult, consider reducing the number of samples in the benchmark.

2. [Quality] While validation metrics are reported, the paper lacks direct human benchmarking, e.g., RM performance vs. human annotators on identical tasks.

3. [Significance, Quality] All tested models are small (-mini/-flash or <10B), which affects the generality of the conclusions when extending to modern large-sized models. For instance, the 25% gap in accuracy may simply be mitigated by increasing model size. To address this concern, experiments on larger models are needed, such as Gemini-2.5-Pro, GPT-4.1/5, or DeepSeek-R1 (671B).

4. [Novelty] Several existing works [1][2] also evaluate LLM steerability. It is worth comparing, or at least discussing, the main differences between EVALUESTEER and these works.

5. [Clarity, Quality]
    * It is strongly recommended to include a figure illustrating the overall synthesis pipeline.
    * The font size in Figure 3 should be enlarged, especially the text in the right subfigure.
    * Table 2: "pm" -> $\pm$
    * Figure 9: Some numbers extend outside the box.
    * Line 1112: ">=" -> "$\ge$"

### References
[1]: Chen, Kai, et al. "STEER-BENCH: A Benchmark for Evaluating the Steerability of Large Language Models." EMNLP 2025.

[2]: Chang, Trenton, et al. "Measuring steerability in large language models." Neurips Safe Generative AI Workshop 2024. 2024.

**Questions:**

* I am wondering whether this style bias can be mitigated as model size increases, similar to a "scaling law" in model steerability.
* I am wondering which version of GPT-4o is used.
* In Section D, a small portion of the synthesized data is manually validated. I am curious about the background of the annotators (the PhD and MS students), in particular:
  - 1) Whether they are experts familiar with different inter-country values.
  - 2) Whether they are authors of the paper and may have biases or conflicts of interest in the rating results.

**Details Of Ethics Concerns:**

* The benchmark is synthesized under different intercountry values, including religious beliefs, whose quality needs to be manually ensured before official release or usage.

---

### Note · Authors · 2025-12-02

**Comment:**

We thank the reviewers for their time and for offering thoughtful and constructive feedback. We have decided, after careful consideration, to withdraw the submission to conduct further revisions and strengthen the work.

**Withdrawal Confirmation:**

I have read and agree with the venue's withdrawal policy on behalf of myself and my co-authors.